# Sex ratios and union formation in the historical population of the St. Lawrence Valley

**Andreas Filser** [1,2]*, **Kai P. Willführ** [1,3]

**1** Institute for Social Sciences, University of Oldenburg, Oldenburg, Germany, **2** Institute for Employment Research, Nuremberg, Germany, **3** Center for Economic Demography, Department of Economic History, Lund University, Lund, Sweden

* andreas.filser@uol.de

## Abstract

The ratio of men and women in the adult population, i.e. sex ratio, has long been recognized as a key demographic constraint for partnering behavior. However, the literature remains contradictory on how sex ratio imbalances influence partnering behavior, suggesting either higher or lower rates of male marriage being associated with male-skewed sex ratios. These contradictory findings are likely due to data limitations. Cross-sectional data or limited observation periods preclude studies from distinguishing sex ratio effects on timing from effects on the overall likelihood of marriage. In this paper, we use historical family reconstitution data to study the association of sex ratios with marriage patterns in the French colony of the St. Lawrence Valley in North America (1680–1750). The population experienced a substantial male-skew from sex-selective immigration during the early period of the colony. The long-running observation period allow for differentiating the timing and overall likelihood of marriage. Finally, the data enable us to study the effects of male-skews on the population-level as well as the regional and parish level. Cox proportional hazard models reveal that while male-skewed sex ratios are associated earlier marriage for women, the association with men's marital biographies is less clear-cut. We find that men marry later when sex ratios are more male-skewed, yet we do not find a substantial reduction in the overall likelihood of marriage for men. Our findings reveal that male-skewed sex ratios do not necessarily result in an increase of never married men. We discuss the implications of our findings for the sex ratio literature.

## Introduction

The ratio of men and women in the adult population, i.e. the adult sex ratio, has long been recognized as a key demographic constraint for partnering behavior [1–8]. However, the literature so far remains inconclusive on how sex ratio imbalances are associated with partnering behavior. Generally, a supply-and-demand perspective suggests that when one sex outnumbers the other, the more numerous sex faces tougher competition for a partner. Thus, individuals of

**Data Availability Statement:** The data underlying the results presented in the study are available from the PRDH project upon request. To obtain the data refer to the following contact details: https://www.prdh-igd.com/en/contact. Moreover, we

provide the full code to replicate our analysis in an OSF repository at https://osf.io/3ejs8/.

**Funding:** The authors received no specific funding for this work.

**Competing interests:** The authors have declared that no competing interests exist.

the scarcer sex are predicted to find a partner and also realize their partner preferences more easily as they have numerous potential mates available [4, 9]. A number of studies support this notion as they report that women are more likely to form unions when they have plenty of potential partners available in a male-skewed sex ratio environment [8, 10–12]. For men, some analyses report that men are more likely to marry when women are more numerous than men [6, 13, 14]. However, other studies suggest the opposite association as they report that men are more likely to marry when men outnumber women [15, 16]. The literature continues to disagree on the explanations for these mismatching findings on sex ratio effects on union formation for men and women [7, 16]. A key factor for the inconclusive findings on sex ratio consequences for marriage and union formation could be the lack of complete marital biographies in existing analyses. Specifically, many sex ratio studies analyzed cross-sectional data or longitudinal data for limited time intervals of 10 years [8, 11, 12, 16]. Such data have only limited potential to differentiate whether skewed sex ratios are associated with shifts in the timing or the overall likelihood of finding a spouse. Specifically, individuals might either find a spouse after the observation period or remain without a spouse permanently. To distinguish postponed from foregone marriages, long-running longitudinal data, ideally on complete biographies, is necessary.

The complete biographies in the family reconstitution data for the 17th and 18th St. Lawrence Valley population present such a unique opportunity to study the consequences of a male-skewed sex ratio on the timing and likelihood of marriage. The complete marital biographies in these data help elucidate whether sex ratio skews shift the timing or rather the likelihood of marriage. Moreover, the specific circumstances of the study population also allow for studying the consequences of a male-skewed sex ratio for marital biographies in an semi-enclosed population [17]. The male-skewed population in the St. Lawrence Valley during the 17th and 18th century is a typical example for sex-selective immigration of European settlers in North America [17, 18].

Regional sex ratio skews emerge in various contexts, which each have specific underlying origins: imbalanced sex ratios at birth and sex-specific infanticide or child neglect, sex-specific mortality disparities, or sex-selective migration [13, 19, 20]. Adult sex ratio skews due to imbalanced sex ratios at birth are typically found in societal contexts with a strong cultural preference for sons. A strong preference for sons has led to the abandonment and neglect of girls in various times and places, including sex-selective abortions in contemporary Azerbaijan, China, or India [21]. Another source of variation in adult sex ratios are differential migration patterns of men and women. For instance in contemporary Europe, populations in rural peripheries are often male-skewed, as women leave these areas more often than men for education and career opportunities in urban centers [20, 22]. Mortality-related adult skew ratio skews are a typical consequence of war, for instance the lack of men in marriageable age in post-WWI France [13]. In either of these cases, the circumstances causing the sex ratio skew also entail consequences for marital behavior. Regional adult sex ratios might rather be an indicator of gender discrimination, economic prosperity, or the impact of war than an isolated cause for marriages squeezes. Studies thus have to distinguish whether regional differences in union formation are attributable to local adult sex ratios or other, underlying contextual factors that also cause local sex ratio skews. In sum, adult sex ratio skews and union formation patterns might be endogenous: instead of one causing the other, both might share the same underlying causes. The male-skewed adult sex ratio in the 17th and 18th century French colony along the St. Lawrence River presents a unique opportunity to circumvent such problems.

The male-skewed adult sex ratio in the 17th and 18th century French colony along the St. Lawrence River presents a unique opportunity to circumvent such problems of endogeneity regarding cultural contexts. Early immigrants to the colony largely were unmarried men,

many of whom came to the colony as soldiers and civil servants. This created a substantial male-skew in the population, which exceeded even those imbalances in contemporary Asian societies, that have received special attention in the literature [19, 21, 23, 24]. Between 1640 and 1670, the unmarried adult population consisted of up to fourteen times as many men than women [18]. The halt of immigration in the 1670s created a semi-closed population with negligible migration from and to the colony [17]. It took a massive population growth from natural fertility for the sex ratio to level off 150 years later. For more than a century, marriage markets in the French colony on the banks of the St. Lawrence River remained male-skewed. During that time, socio-economic and cultural contexts largely remained the same. Socioeconomic differences were modest for the period under observation, as the vast majority of inhabitants were subsistence farmers [25].

In this paper, we leverage these exceptional circumstances of the male-skewed population in St. Lawrence Valley colony to study marriage market dynamics both on the level of the colonial as well as on smaller scaler, regional levels. Unlike regional adult sex ratio skews from sex-selective migration in contemporary Europe which imbalance sex ratios mainly on regional levels, the entire population of French origin in the St. Lawrence Valley was affected. While the area was settled by native North Americans, intermarriage between French-origin settlers was limited to few exceptional cases. Therefore, settlers did have not a meaningful alternative option to find a partner in a region close-by, where sex ratios might be more favorable. Instead, the St Lawrence Valley colony constituted a semi-closed population, as individuals only emigrated from the colony during select periods and under specific circumstances [17]. This allows us to analyze consequences of local and colony level sex ratios on marital outcomes in an encapsulated environment. In addition, the long-running, multi-generational scope of our data allows us to study complete marital biographies. Previous studies using cross-sectional or longitudinal data for shorter time periods cannot distinguish between postponement and likelihood of marriage. Here, we are able to disentangle potential associations of sex ratios with the timing and likelihood of marriage for men and women. Moreover, the family reconstitution data also allow us to adjust our models for unobserved heterogeneity shared between families or places of birth. This presents an outstanding opportunity to explore how sex ratio effects might be cofounded by systematic differences between families or places of birth. Finally, we use geographic information on residence to approximate individuals' contextual sex ratios of neighboring parishes based on distances to model local marriage markets more adequately. Thus, we are able to circumvent problems of too high levels of aggregation when calculating sex ratios, which have been identified as a major source for mixed findings on the link between sex ratios and transitions into marriage [8, 26–29].

## Materials & methods

### Data

The data for our study come from the *Registre de population du Québec ancien* (RPQA), compiled by the *Programme de recherche en démographie historique* (PRDH) at the Université de Montréal [17]. RPQA is a family reconstitution database that comprises the population that lived along a 500 km stretch on both sides of the St. Lawrence River. In total, RPQA contains approximately 700,000 baptismal, marriage and burial acts dated from 1621 to 1799. Burial acts of individuals who died until 1850 have been added to the database and linked to acts preceding 1800, allowing observation of life courses of individuals born before 1750 [17]. Colonial census data, marriage contracts, hospital sick lists, and lists of migrants complement the data base. Record losses are relatively few and missing information primarily concerns infants dying before baptism, young children, and people who died outside of the parish [17, 30–32].

The particular historical and geographical circumstances of the settlement of the St. Lawrence Valley created a 'semi-closed population' until the 19th century, so the usual problem of missing observations resulting from emigration is substantially reduced [17, 33]. Moreover, loss of individuals due to inter-parish migrations is marginal given that the RPQA data cover all parishes of the colony and records for individuals who migrate between parishes have been linked. In sum, the RPQA data permit an exceptionally close reconstitution of complete individual biographies.

The data allow to reconstruct the sex ratio context of the colony over time on the parish level (see below), enabling us to estimate the association of sex ratios with the probability and timing of marriage, respectively. Our sex ratio estimates include persons who immigrated to the colony from France as potential partners and spouses on the marriage market. However, we exclude the marital biographies of immigrants from our statistical models. During the early period, immigrants account for the majority of individuals in the colony and the immigrant population was more male-skewed than the Quebec-born population [17, 18]. The inclusion of this (self-)selected group who left France and who survived the passage to North America would result in a survivorship bias. Furthermore, data on immigrants, such as date of birth or previous marriages, are often incomplete, precluding statistical analyses. Our analysis therefore focuses on marital biographies of individuals born in the colony.

## Historical context and observation period

During our study period, St. Lawrence Valley colony was a frontier society, a population deriving from a small initial influx of immigrants [34, 35]. Most of these immigrants came from France (95 percent) [36]. Following the foundation of Québec City in 1608, early immigrants were mainly men who came to the colony as soldiers, who were offered incentives to stay in the colony. Others came as indentured servants or were smugglers or other minor offenders deported by the crown [37]. Given that of all the migrants, only the indentured workers chose to migrate to the colony, the population has been described as reluctant migrants [38]. Most migrants either came to the colony involuntarily or had to be heavily subsidized to stay, given that the St. Lawrence valley offered only a very limited living-standards premium compared to native France [39, 40]. In contrast to prospering British and also Spanish colonies on the American continent, this lack of immigration premium prevented a sustained immigration influx from France [38–40]. Moreover, the small share of indenture servitude among the male immigrants provided little cause to create a voluntary immigration influx of women that might follow from indentured servants migration, for instance in the American colonies [41]. This resulted in a male-skewed population sex ratio, as men outnumbered women by a factor of up to 14 in the 1650s [18]. To stimulate population growth and incentivize more of the male immigrants to stay in the colony, the French crown sponsored a female immigration program. In total, 716 French women came to the colony as part of the *filles du roi* ("King's daughters") program between 1663 and 1673 [42]. The arrival of these women started an exponential population growth through natural fertility. Studies report an average of almost 10 children per married woman reaching the age of 50 [25, 43, 44]. While the colonial population comprised only 3,200 individuals at first census taken in the colony in 1666, it multiplied to a size of 70,000 by 1760 [32]. Although the population growth from natural fertility counter-balanced the male-skew of the colonial sex ratio, men continued to outnumber women in the colony until around 1750.

We limit our observation to the period to between 1680 and 1750. This corresponds to the period between after the end of *filles du roi* immigration program and before the French and Indian war [35]. The war increased the immigration of French settlers from the southern New

France territories lost to the British. In particular, the British Conquest of the colony was followed by a significant emigration to France between 1755 and 1770 [17]. Given that these emigrants might be at least partially unobserved in the RPQA data, we exclude the war period from our study to prevent bias in our sex ratio estimates. In total, our analytical sample contains 44,252 individuals deriving from 93 parishes within the St. Lawrence Valley.

Socioeconomic differences were modest for the period under observation. The observation period ends before the population underwent industrialization [17, 45]. The vast majority of inhabitants lived outside the urban regions of Montréal and Québec City as subsistence farmers [25]. However, resource availability varied by geographic location, as the southern shore of the Saint-Lawrence River provides a higher potential for agriculture than the northern region. This is also illustrated by differences in life expectancy between the northern and southern parishes [25].

## Estimation of sex ratios

Our key independent variable of interest is the sex ratio (SR), operationalized as the share of men in the unmarried (including widowed) population aged 14 to 50 who were present in the colony in a given year. Thus, our SR measure fits the definition of an operational SR, which captures the number of men and women that are available to potential partners. For comparison, we also estimate SRs based on the entire population, including married individuals. For either variant, we operationalize the sex ratio as the percentage share of men to avoid the asymmetry of proper ratios [46].

Based on this definition, we calculate three sex ratio measure variants, which differ in their geographic scope. Specifically, we estimate a colonial, regional and parish-level sex ratio measure. The colonial SR is an estimate of the share of men in the colonial unmarried population. For the parish-level SR, we calculate the share of men separately for each parish. The regional SR reflects the share of men in the unmarried population of the focal parish and the closest neighboring parishes. The neighboring parishes are determined based on the aerial (Euclidian) distances between the parishes [47, 48]. For each parish, we selected the five closest parishes for our regional sex ratio measure. Moreover, we added further neighboring parishes if there were any other parishes up to 500m farther away from the focal parish than the fifth closest parish. Consequently, the regional SRs include the six closest parishes for 15 focal parishes and seven neighboring parishes for two more parishes out of the 114 parishes in total. For the remaining parishes, the regional SR is calculated based on the focal and the five closest parishes by aerial distance.

## Localization

Given that our sex ratio measures cannot rely on population records but reconstructions from individual-level parish records, we need to localize individuals in the periods between these records. Localization of individuals is an important aspect of our study in two respects. First, we need to approximate when immigrants from France enter the colonial marriage market to approximate the colonial SR in a given year. Second, we need to localize individuals in specific parishes within the colony to estimate parish-level and regional SRs.

We localize individuals within the colony based on the parish records, which provide information where the events took place. However, individuals usually do not appear in the data between their birth and their first marriage. This gap creates ambiguity with respect to the place of residence for the period between records, particularly shortly prior to marriage. We fill this gap based on the birth certificates of younger siblings, assuming that children lived with their parents until the age of 14. In sum, this information yields reasonable certainty for

the place of residence for individuals whose records all come from the same parish. Individuals most likely stayed in the same parish if all records of their own and their siblings' baptisms as well as their marriage record come from the same parish. The localization of individuals with records from different parishes is more challenging. Previous studies estimate that a fifth of grooms married in a different parish than their place of residence–typically their bride's parish–without necessarily moving to this parish [49]. Therefore, it would be inadequate to conclude that a move has occurred each time an individual marries outside his or her parish of birth [17, 49]. While this constitutes a challenge for inferences of the exact place of residence, it is less of a problem for our purpose of reconstructing local marriage markets. The fact that the individuals in question married one another demonstrates that they were part of the same marriage market at some point. Therefore, approaches localizing individuals prior to marriage in their parish of residence or their parish of marriage both have analytical advantages. Given that neither approach is absolutely preferable over the other, our analysis comprises sex ratio measures from both localization approaches.

A similar ambiguity arises with respect to the date of immigration for French-born individuals. While exact dates of immigration are unavailable, retrospective records of demographic events in France help to narrow down the time window of arrival at the colony. Nevertheless, we cannot fully determine when individuals enter the colonial marriage market. At the same time, estimating immigration dates is crucial to arrive at accurate sex ratio estimates, because in the early period, immigrants account for the majority of individuals on the marriage market. Moreover, the immigrant population was more male-skewed than the Quebec-born population. Therefore, excluding immigrants from the sex ratio estimation would result in a substantial underestimation of the ratio between men and women.

We address these ambiguities using two different strategies to localize individuals outside or and within the colony. First, we apply a retrospective localization strategy that places individuals in their parish of birth until a conflicting birth record of a sibling or a marriage record suggests a different location (Fig 1, second timeline). A second strategy takes a prospective perspective: here, we consider a demographic event as the end of a residence period in a parish, if the subsequent record documents a different parish (Fig 1, third timeline). In sum, the retrospective strategy yields a conservative estimate of the latest plausible arrival of an individual in the colony and a specific parish. The prospective strategy on the other hand gives an estimate based on the earliest plausible arrival in the colony and a specific parish.

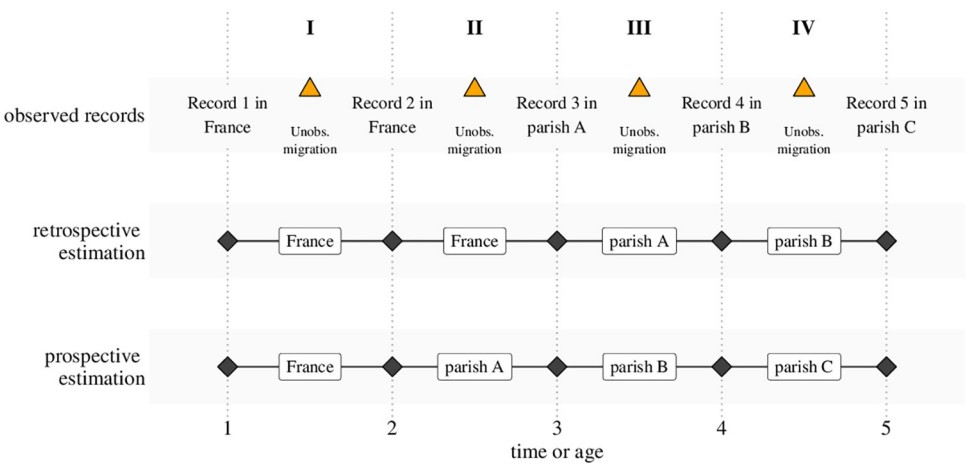

**Fig 1. Different variants of sex ratio estimation.**

Note that some records do not provide information on the exact locality of the demographic event. For instances, the bride or the groom may report that they were born somewhere in Quebec (the colony) without any further details and no further record exists to identify their place of birth. Individuals with such records are included in the colonial sex ratio estimates. However, these episodes are not included in the regional and parish-level sex ratio estimates, given that they cannot be assigned to a parish. The regional and local sex ratio estimates therefore are calculated from slightly smaller samples than the colonial sex ratio. Specifically, the retrospective smaller-scale sex ratio estimates rely on 3.1% (48,938) fewer episodes than the colonial estimates. For the prospective approach, 4.1% (71,424) of the episodes used for the calculation of the colonial sex ratio are not usable for the regional and local sex ratio estimates. Finally, we exclude opposite-sex siblings from the sex ratio in order to include only eligible individuals as potential partners.

## Analytical approach & control variables

We explore the association of transitions into marriage with our different sex ratio estimates using Cox proportional hazard models. Models are fitted based on life courses for individuals from sexual maturity (14 years of age) to age at first marriage, censored at age 50. In order to accommodate the time-varying, annual sex ratio estimates, we split observation episodes at June 15th of a given year. Given that we localize individuals based on the year of a parish record, episode-splitting at the mid-point of one year is more appropriate the splitting at January 1$^{st}$. Our analysis includes individuals who are born in the colony, and we restrict our analytical sample to the period between 1670 and 1749. During this period, the settlers in the St. Lawrence Valley formed a semi-enclosed population with only negligible migration from and to the colony [17]. Individuals for whom one or more marriage dates are unknown are excluded from the analysis. We estimate separate model series for women's and men's life courses.

For our analysis, we estimate separate models with different specifications in terms of sex ratio measures, localization strategies, control variables, and fixed effects. Specifically, we analyze the association of each colonial, regional, and parish-level SR with the transition to marriage in separate model series. Each series consists of models using different combinations of control variables and fixed-effects, respectively. The complete list of specification variants is summarized in Table 1. These versions of model series are estimated both using the retrospective and prospective localization approach (see section 2.4).

The respective sex ratio measure is modelled as a categorical variable to allow for non-linear relationships and to alleviate potential multi-collinearity problems from the time trend in the colonial SR. Specifically, we categorize sex ratios into (1) <45%, (2) 45 <49%, (3) 50 <54%, (4) 55<59%, (5) 60 <64%, and (6) >65%. In all models, (4) 55<59% is set as reference category. As indicated by the descriptive results (see section 3.1.1), the colony-level SR follows a clear time trend from a male-skewed towards a more balanced sex ratio. Model variants 2 and 4 therefore adjust for period effects by including the decade of the episode start as a time-varying categorical covariate. In model variants 3 and 4, we adjust for observed family characteristics based on birth order, number of siblings, number of brothers and sisters alive at age 14, and maternal as well as paternal loss. Moreover, we control for unobserved characteristics by fitting different fixed-effects versions for each model variant. Specifically, models with mother-fixed-effects control for unobserved heterogeneity shared between siblings, parish-fixed-effects adjust for unobserved shared characteristics of individuals residing in the same parish, and birthplace-fixed-effects to control for unobserved characteristics which derive from conditions in the birthplace.

**Table 1. Model variants.**

| Model | Control variables | Fixed-effects | | |
|---|---|---|---|---|
| | | Mother | Parish | Birthplace |
| 1 (basic model) | None | | | |
| 1-sfe | None | x | | |
| 1-pfe | None | | x | |
| 1-bfe | None | | | x |
| 2 | Decade (to control for period effects) | | | |
| 2-sfe | Same as 2 | x | | |
| 2-pfe | Same as 2 | | x | |
| 2-bfe | Same as 2 | | | x |
| 3 | Birth order, number of siblings, number of brothers and sisters alive at age 14, maternal as well as paternal loss (both time-varying) | | | |
| 3-sfe | Same as 3 | x | | |
| 3-pfe | Same as 3 | | x | |
| 3-bfe | Same as 3 | | | x |
| 4 (full model) | Decade, birth order, number of siblings, number of brothers and sisters alive at age 14, maternal as well as paternal loss (both time-varying) | | | |
| 4-sfe | Same as 4 | x | | |
| 4-pfe | Same as 4 | | x | |
| 4-bfe | Same as 4 | | | x |

All analyses and visualizations are performed using R 4.1.2 and the tidyverse and data. Table packages [50–52]. We provide the full code to replicate our analysis in an OSF repository at https://osf.io/3ejs8/ [53]. The PRDH data is freely available to the scientific community at https://www.prdh-igd.com/en/contact.

## Results

### Descriptive results

**Development of sex ratios of over time.** Fig 2 displays the SR in the population aged 14–50 of the St. Lawrence Valley between 1680 and 1750 based on our different localization strategies. In general, we find that the colony-level SR was markedly male-skewed during the 17th century and declined steadily throughout our observation period. While the overall pattern is visible in all of our SR measures, the extent of male-skew differs depending on which individuals are included. Including both married and unmarried individuals yields more balanced SRs than when focusing on the unmarried population only. This is consistent across all three localization methods. In fact, the retrospective and semi-prospective localization methods yield the same estimates for the colonial sex ratios. However, SRs based on the retrospective localization differs substantially from the SR estimates based on a prospective localization. This difference is due to the differential handling of immigrants from France. While the prospective approach includes immigrants at the earliest possible time point, the retrospective approach only includes immigrants at the latest possible time. Given that immigrants were largely men, the SR variants based on the prospective approach yield more male-skewed SRs estimates than the retrospective approach. We consider these estimates a lower and upper bound for the colonial sex ratio.

**Transition into first marriage.** Fig 3 displays the colonial SR for the unmarried population as well as the male and female ages at first marriage for individuals born in the colony. In

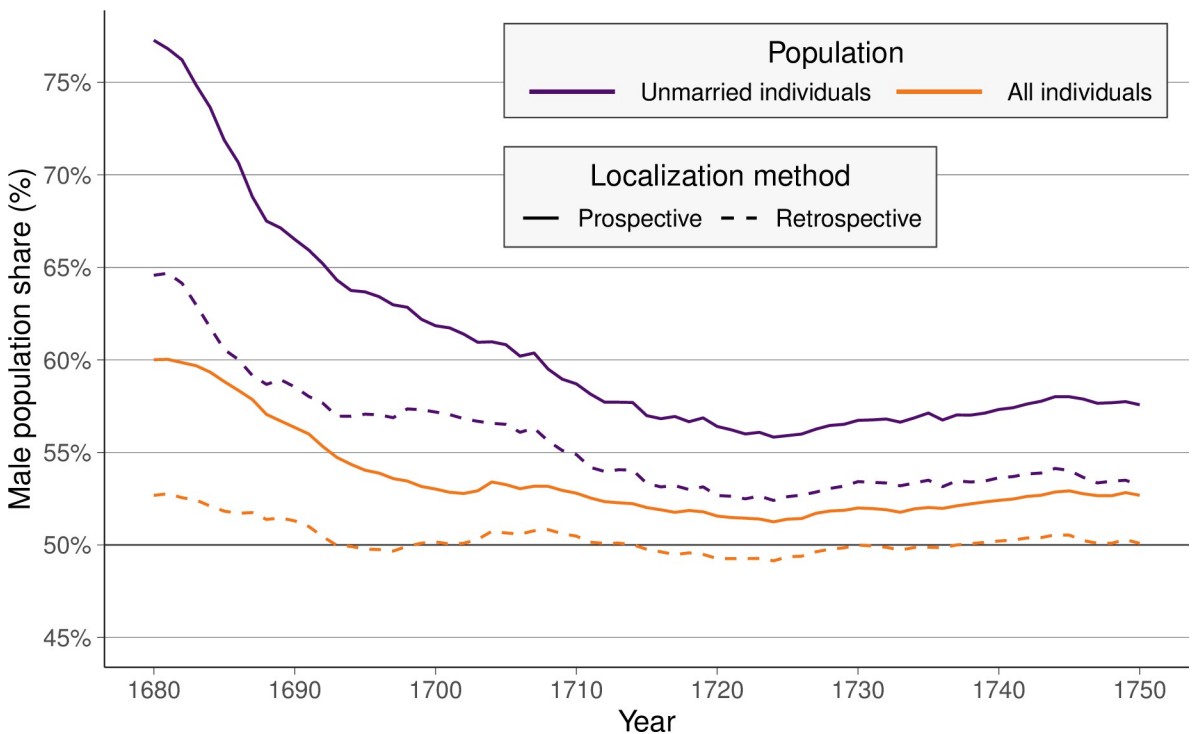

**Fig 2. Different sex ratio measures for the population of the St. Lawrence valley.** Individuals aged 14–50. PRDH data, 1680–1750.

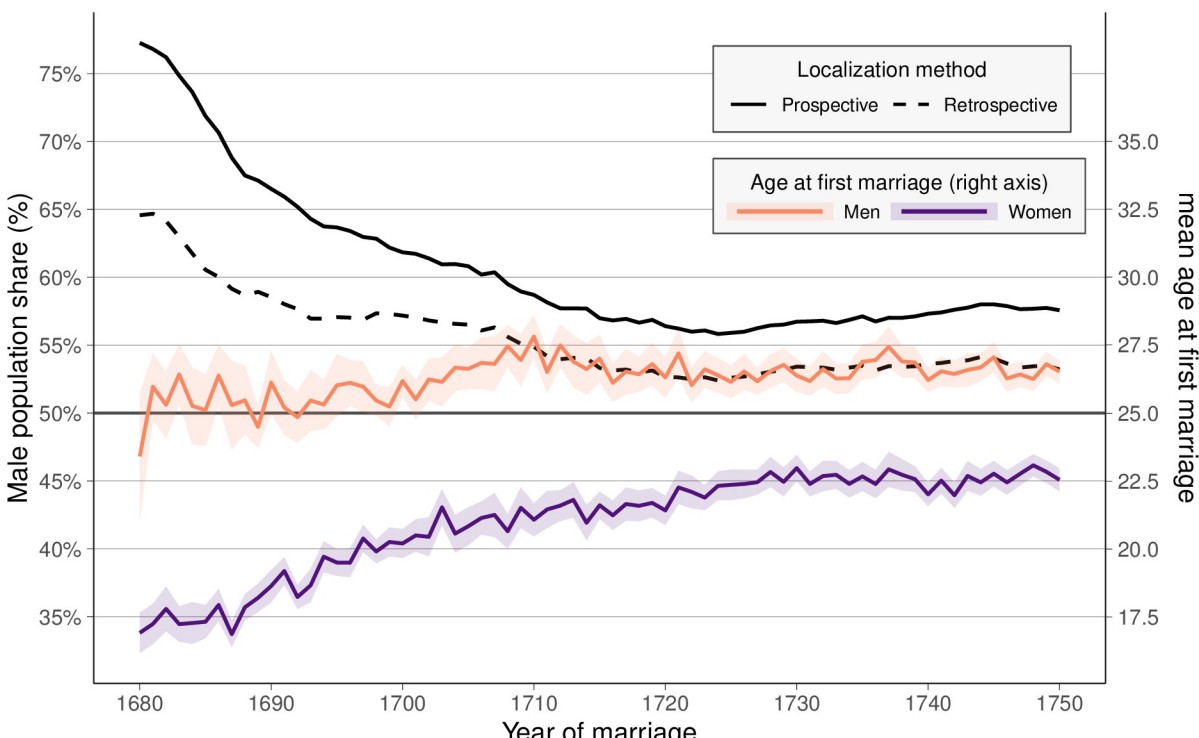

Note: Colored ribbons around ages at first marriage indicate 95 percent confidence intervals

**Fig 3. Share of men in the unmarried population and mean age at first marriage by sex.** PRDH data, 1680–1750.

**Table 2. Likelihood of marriage for individuals born in the St. Lawrence Valley parishes by birth cohort.**

| birth cohort | Married before 45 | Married over 45 | Died unmarried over 45 | Died unmarried before 45 | Married, unknown age | N | |
|---|---|---|---|---|---|---|---|
| 1660 | 90.3 | 0.2 | 5.7 | 3.8 | 0.0 | 631 | **Women** |
| 1670 | 87.8 | 0.3 | 4.9 | 7.0 | 0.0 | 1,489 | |
| 1680 | 85.2 | 0.5 | 6.3 | 8.1 | 0.0 | 1,495 | |
| 1690 | 83.1 | 0.3 | 7.2 | 9.4 | 0.0 | 2,326 | |
| 1700 | 85.7 | 0.5 | 4.6 | 9.2 | 0.0 | 3,134 | |
| 1710 | 84.8 | 0.6 | 4.2 | 10.4 | 0.0 | 3,747 | |
| 1720 | 85.9 | 0.3 | 4.2 | 9.6 | 0.0 | 4,898 | |
| 1730 | 85.0 | 0.5 | 4.0 | 9.8 | 0.6 | 6,573 | |
| 1740 | 86.2 | 0.5 | 3.9 | 7.2 | 2.2 | 6,909 | |
| 1660 | 79.3 | 0.8 | 3.5 | 16.4 | 0.0 | 593 | **Men** |
| 1670 | 77.8 | 2.0 | 4.5 | 15.7 | 0.0 | 1,223 | |
| 1680 | 78.4 | 1.2 | 4.8 | 15.6 | 0.0 | 1,322 | |
| 1690 | 80.8 | 1.0 | 5.2 | 12.9 | 0.0 | 2,040 | |
| 1700 | 81.2 | 0.8 | 4.0 | 14.1 | 0.0 | 2,807 | |
| 1710 | 80.6 | 1.0 | 4.4 | 14.0 | 0.0 | 3,413 | |
| 1720 | 81.9 | 0.8 | 3.7 | 13.5 | 0.2 | 4,438 | |
| 1730 | 77.9 | 0.6 | 4.0 | 15.3 | 2.2 | 5,800 | |
| 1740 | 80.1 | 0.6 | 4.1 | 9.8 | 5.5 | 6,391 | |

Values are percentages. Source: PRDH data, own calculations

sum, the average female age of first marriage increases as the number of excess men on the colonial marriage market declines. Initially, when the SR is heavily male-skewed, brides are around 17.5 years old, while women are around 22.5 years at marriage at a less a male-skewed sex ratio in the 1750s. However, male ages at first marriage do not exhibit a similar clear trend in average ages at first marriage. While there is some year-to-year variation, there is only a small overall trend with men being around 25 years old at their first marriage in the 1680s, compared to around 26 years in the 1750s. In sum, marital age hypergamy declines steadily and converges towards age differences that are typical in most populations. In 1680–82, grooms are 7.6 years older than their brides are, whereas newlywed husbands are on average only 3.7 years older than their wives in 1748–50.

The SR might not only correlate with the timing, but also with the proportion of individuals marrying. However, Table 2 reveals that unlike the ages at first marriage, there is no trend in individuals' likelihood of marriage across cohorts for neither men nor women. Specifically, the share of never married men and women surviving until 45 only varies idiosyncratically throughout the observation period. Contrasting this essentially flat curve with the trend in the colonial SR suggests no association between the two. Moreover, this share of never married individuals constantly remains under 5 percent (men) and 7 percent (women), respectively. In other words, we do not find a substantial *cure* fraction of individuals who died unmarried despite they lived until their adulthood.

Consequently, cox proportional hazard models are suitable for modelling the association of individual-level transitions into first unions with local sex ratios. Specifically, cox proportional hazard models rest on the underlying assumption that the event of interest–marriage in our case–will eventually occur if there is no censoring [54, 55]. The very small cure fraction of unmarried individuals illustrates that this assumption holds for the data analyzed here.

## Multivariate results

Moreover, we fit multivariate models to analyze whether the patterns in the descriptive analyses persist if we adjust for period effects and a number of individual-level controls. Fig 4 displays the hazard ratios from a series of Cox proportional hazard models predicting transition into first marriage for men and women (the full regression tables are available in S1 Table). The upper half of Fig 4 summarizes the results based on the retrospective localization

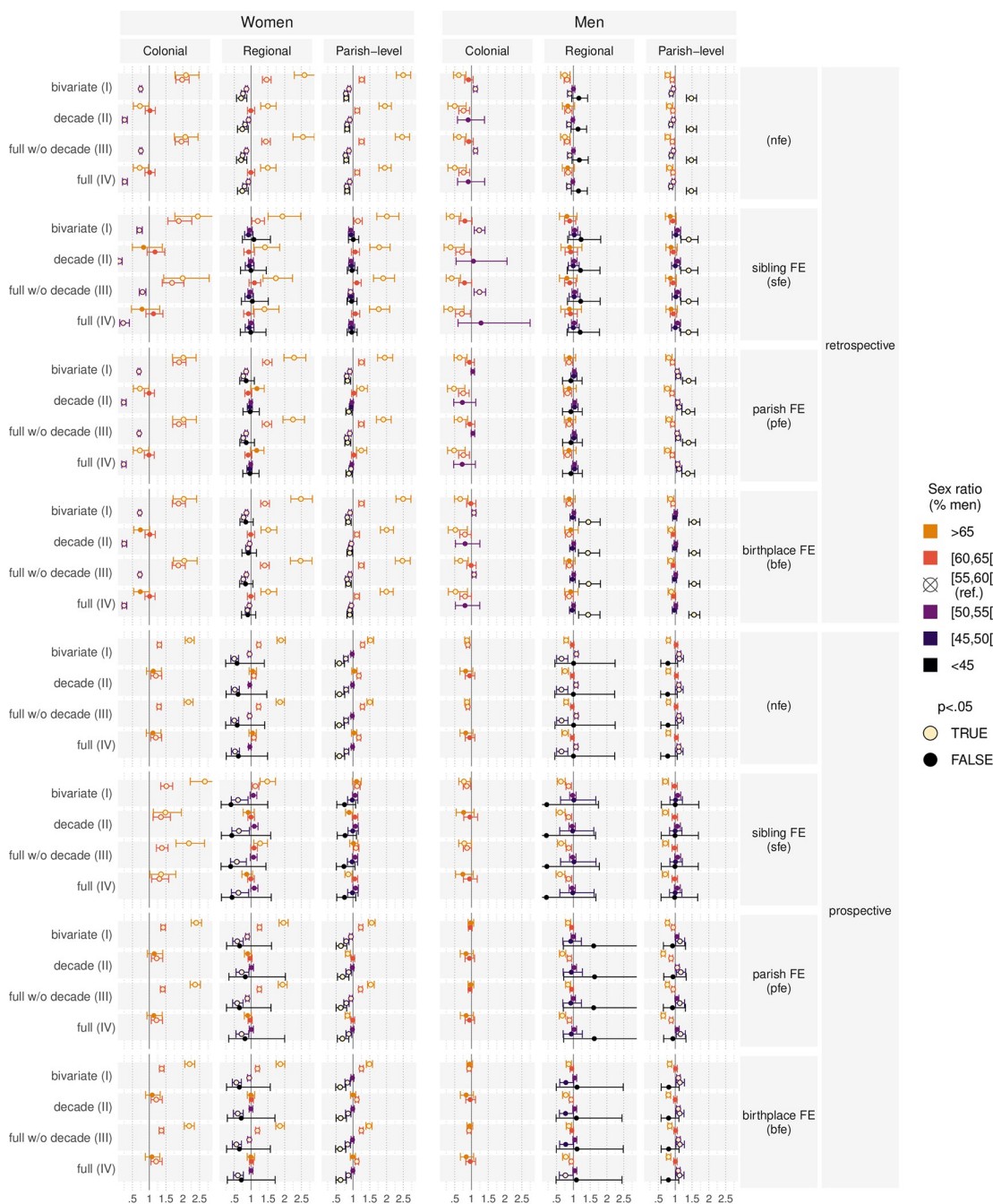

**Fig 4. Hazard ratios from Cox proportional hazard models predicting transitions into first marriage.**

approach, while the bottom half displays the coefficients for sex ratio measures based on the prospective localization approach (see section 2.4).

**Sex ratios and female transition into marriage.** The model variants III without the decade controls suggest that women marry earlier when there are more men on the marriage market. This statistically significant finding is consistent in both model series using the retrospective and the prospective localization approach as well as on the colonial, regional, and parish level. When the colonial, regional, and parish-level SRs are at 60–65% or greater than 65%, women marry earlier than the reference category (55–60%). SRs below the reference category correlate with later first marriage for women.

Models controlling for the decade (IV) yield a less clear relationship between the colonial SR and female age at first marriage. Model variants IV without fixed effects or parish-fixed-effects models yield a u-shaped relationship when employing the retrospective localization approach. Compared to the reference category, women marry later both when the SR is greater than 65% or between 50–55%. However, the corresponding period-adjusted models using the prospective localization approach suggest a different relationship. Here, results suggest that women married statistically significantly earlier than the reference when the colonial SR was 60–65%. Similarly, model versions of IV using sibling-fixed-effects also suggest that women married earlier when the colonial share of men was 60–65%. Moreover, sibling-fixed effects models also suggest that women marry significantly earlier when the colonial share of men was greater than 65%. However, the coefficient for SR > 65% is not significant of the other model configurations relying on the prospective localization approach.

The results for the regional and the parish-level SR from period-adjusted models (IV) are consistent across localization approaches. Period-adjusted models without fixed-effects suggest that regional and local shares of men smaller than the reference category are associated with later first marriage. This finding is consistent for both localization approaches, albeit the prospective localization yields a statistically insignificant coefficient for a SR below 45%. Birthplace-fixed-effects confirm findings from models without fixed effects for both localization approaches. Yet, introducing sibling- or parish-fixed-effects into the period-adjusted models attenuates the hazard ratios into insignificance for both localization approaches. Generally, coefficients for parish-level and regional SRs from sibling-fixed effects are even less pronounced than those in parish-fixed effects models across both approaches. Yet, results differ across localization approaches for regional and parish-level SR that are greater than the reference category. Based on the retrospective localization approach, period-adjusted models indicate that women married earlier when the regional or parish-level SR were greater than 65%. The corresponding models using the prospective localization approach do not suggest that a SR greater of 65% is associated with lower age at marriage.

In sum, results from Cox regression models suggest that women married earlier when the SR was more male-skewed. This association, however, is superposed by a time trend which limits the potential to disentangle period effects and SR effects. Moreover, we find that sibling- and parish-fixed-effects change the results differently across the localization approaches.

**Sex ratios and male transition into marriage.** For men, almost all models which use the retrospective localization approach suggest that men marry statistically significantly later when the colonial SR is more male-skewed. When the colonial SR is 60–65% or greater than 65%, men marry later in comparison to the reference category (55–60%). Moreover, models not adjusting for decade (III) suggest further that men marry earlier than the reference when the colonial SR is 50–55%. The only exception of models with parish-fixed-effects.

However, models using prospective localization approach do not suggest a similar association. None of the decade-adjusted models (IV) suggests a statistically significant association between the SR categories and male age at first marriage. Only selected models without decade

adjustment (III) suggest a decelerating effect of the colonial SR on men's transition into marriage. The parish-fixed-effect model versions do not yield any significant hazard ratios for the SR categories.

Results for regional and parish-level SRs are more consistent across the retrospective and the prospective localization. Results suggest that who married later in life compared to the reference when the regional or the parish-level SR exceeds 65%. Sibling-fixed-models using the retrospective localization approach are the only exception from this general finding. Most of the models further reveal that men are also older at first marriage when the regional or the parish-level SR is 60–65%. For regional and parish-level SRs below the reference, fixed effects model series for both localization approaches indicate that men are either statistically significantly younger or yield no statistically significant difference relative to the reference category.

However, models without fixed-effects based on the prospective localization indicate a U-shaped relationship between the regional SR and male age at first marriage. Specifically, they yield that men marry statistically significantly later compared to the reference when the regional SR is 50–55% or 45–50%. Similarly, the models without fixed-effects using the prospective localization approach which estimate the effect of regional SR reveal that men are older when the SR is 45–50%. The model for the retrospective parish-level SR without fixed effects also suggests earlier marriage for men when the parish-level SR is 50–55% or 45–50%. Since all models with fixed-effect imply a linear relationship between the share of men and male age at first marriage, the nonlinear relationship found in models without fixed-effects is most likely caused by unobserved heterogeneity on the family as well as on the parish and birthplace level. Finally, results for the 45–50% SR category should be treated with caution, as it contains a very limited number of observations: 226 individuals in six parishes after 1710 experience a total 1.311 episodes (of 215.531) with a SR of 45–50%.

## Discussion

This study investigates the association of sex ratios with marriage patterns in the French colony of the St. Lawrence Valley in North America (1670–1750). The long-running data on complete biographies and the semi-closed study population allow for comprehensive analyses of the association of male-skewed sex ratios with men's and women's timing and likelihood of marriage. In sum, aggregate descriptive results suggest that male-skewed sex ratio in the colony was associated with earlier female first marriage, whereas the association is less clear for male ages at first marriage. In sum, aggregate descriptive results suggest that male-skewed sex ratio in the colony was associated with earlier female first marriage, whereas the association is less clear for male ages at first marriage. When the colonial sex ratio was heavily male-skewed, women married under the age of 18. As the colonial sex ratio became more balanced, women's median age at first marriage increased to around 22 years in the 1740s. Even these increased ages at first marriage in the St. Lawrence Valley were substantially lower than in the donor population in France, where the median age at first marriage was around 27 for women and 29 for men in the 1740s [56]. Men in the St. Lawrence Valley on average married between age 25 and 27.5 throughout the study period, regardless of the colonial sex ratio. Moreover, sex ratio imbalances were not associated with the fraction of unmarried men at age 45. This finding is closely associated with the high mortality in the study population, which implies that unmarried men frequently died before 45 or had the option to marry a widow after her spouse died.

Multivariate individual-level analyses reveal a more complex picture of the association between male and female timing of first marriage and the colonial sex ratio. A key challenge for our multivariate analysis is the distinct time trend from a male-skewed towards a more balanced colonial-level sex ratio. For female age at first marriage, the multivariate results reveal

that the distinct effect of the colonial sex ratio suggested by the descriptive results even persists when adjusting for the time trend. For men, multivariate results suggest that more male-skewed colonial sex ratios are only partially associated with a later transition into marriage. Specifically, we only find a significant association with the colonial sex ratio measure that includes immigrants at the latest possible date of immigration which is usually their date of marriage. The opposite approach–including individuals before their marriage–reveals that men did not marry significantly earlier when the colonial sex ratio was more male-skewed. For both men and women, the results are consistent when integrating family-, birthplace- or parish- fixed effects. Finally, we also analyzed regional and parish-level sex ratios. Beside the trend in the colonial sex ratio, within-colony variation could have been associated with settlement history. S1 Fig reveals that that newly established parishes tended to have male-skewed sex ratios when prospectively localizing individuals (S1 Fig). This hints at the demand for male labor force to clear the land on the frontier–men moving to newly established parishes before marriage. Our results consistently indicate that men were older and that women were younger at their first marriage when there were more men on the regional and local marriage market. In contrast to the colonial sex ratio, we find that unobserved heterogeneity affects our results for men and women differently. For women's age at first marriage, we find that models including sibling- or parish-fixed-effects yields mostly non-significant estimates for the regional and parish-level sex ratios. This suggests that the association between female age at first marriage and the sex ratio on the regional and parish-level is largely caused by unobserved characteristics shared within the family and the parish of residence, respectively. For men, the role of unobserved heterogeneity on the family and parish level appears to be less important than for women. Almost all models indicate that men married later when they outnumbered women more severely on regional and the parish-level. For men, the most important finding of this study is that men's age at first marriage was increased by male-skewed sex ratios, but that there was no considerable increase in the fraction of men who stayed unmarried. In other words, male-skewed sex-ratios produced longer waiting times for men but did not increase the fraction of older unmarried men. The male-skewed sex ratio is counterbalanced by high levels of mortality to result in only a very small number of surviving, never-married men. The study population was subject to pre-demographic transition levels of high mortality [25, 57]: Men were involved in physically demanding and hazardous activities, including clearing of land, fur trade, or militia duty. The resulting high mortality contributes to our findings in two ways. First, a substantial proportion of men do not survive until age 45, therefore limiting the number of men reaching the age barrier to be counted as never married adults. Second, the high mortality also affects married men, which results in widows re-entering the marriage market and thus increasing surviving men's chances to find a spouse eventually. Therefore, the consequences of imbalanced sex ratios on the share of men who remain unmarried until they turn 45 might be different in a modern, low-mortality environment. Still, our findings highlight the need for longitudinal data when studying the consequences skewed sex ratios on marital patterns. Differential consequences of sex ratio skews for the timing and quantum of marriages can only be revealed in longitudinal data, ideally covering complete demographic biographies. Our findings match results from an analysis of the historical population in Utah between 1880 and 1900 [6]. Results for the Utah population suggest that men's chance to ever marry is not associated with the local adult sex ratio, thus resembling our findings. For women, the Utah study finds that women are more likely to marry when the adult sex ratio is more male-skewed. This is also consistent with our findings, as we find that women marry earlier when sex ratios are more male-skewed.

Nevertheless, the potential of our data comes with some limitations. Specifically, while being able to study marriage patterns and sex ratios at both the population and smaller levels,

even smaller scale sex ratios affected by the clear time trend in the colonial sex ratio. The steady decline of the colonial sex ratio is accompanied with an increasing population and the expansion of infrastructure. Furthermore, the underlying reason for the male-skewed sex ratio–the arrival of predominantly male immigrants in the colony–disappears over time as individuals born in the colony make up an increasing share of the population. We address this problem by including controls for the decade of observation. Yet, this does not fully eradicate the underlying collinearity between the sex ratio and the establishment of the colony. Additionally, while the parish-level and regional sex ratio might be female-skewed in some instances, individuals where still exposed to overall male-skewed sex ratio of the overall colonial population. Thus, our results for female-skewed small-scale sex ratios should be interpreted with caution when generalizing to other female-skewed contexts. However, the unique circumstances in the St. Lawrence Valley and the long-running data on the colonial population allow for an analysis of sex ratio skews that few other sources can provide.

In conclusion, our results might explain the contradictory findings on the association of male-skewed sex ratios with the timing and likelihood of marriage for men. Studies report that men are either more likely to marry when women are abundant while others suggest that men are more likely to marry when men outnumber women [6, 13–16]. These contradictory findings might be partially attributable to mediating effects from norms regulating non-marital sexual involvement [6, 58–60]. However, our analysis shows that while we find that male-skews postpone marriage for men, the link between sex ratios and the chance to marry is not as clear. Overall, we find that men do find a spouse eventually even when women are scarce. This illustrates the need for long-running, longitudinal data to study marital involvement to differentiate postponed and foregone marriages. This is not to dispute the importance of cultural contexts in shaping the consequences of skewed sex ratios on marital outcomes, but analyses of cross-sectional data might be a key source of contradictory findings, given the inability to discern quantum from timing. Specifically, cross-sectional census information for calculating sex ratios cannot capture migration in response to sex ratios skews. For instance, cross-sectional studies fail to capture whether men in disadvantageous marriage markets might migrate to more promising marriage markets. The longitudinal data on the semi-closed population our study allows us to circumvent these problems [17]. Future sex ratio research should seek to analyze data that allows for disentangling the specific consequences on marriage among men. Moreover, studies relying on shorter time periods should be aware of the implications when analyzing such data on phenomena that commonly necessitate long-term data.

## Supporting information

**S1 Table. Full regression tables from Cox proportional hazards models.**
(PDF)

**S1 Fig. Mean sex ratio by parish age.** Parish-level SR estimates for unmarried individuals by localization method and different parish age cut-offs.
(PDF)

## Author Contributions

**Conceptualization:** Andreas Filser, Kai P. Willführ.

**Data curation:** Andreas Filser, Kai P. Willführ.

**Formal analysis:** Kai P. Willführ.

**Investigation:** Andreas Filser.

**Visualization:** Andreas Filser.

**Writing – original draft:** Andreas Filser, Kai P. Willführ.

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
