## [Decision Letter · Decision Letter 0]

7 Feb 2022

PONE-D-21-25452Sex ratios and union formation in the historical population of the St. Lawrence ValleyPLOS ONE

Dear Dr. Filser,

Thank you for submitting your manuscript to PLOS ONE. After careful consideration, we feel that it has merit but does not fully meet PLOS ONE’s publication criteria as it currently stands. Therefore, we invite you to submit a revised version of the manuscript that addresses the points raised during the review process. Both reviewers are quite positive about the contribution of your article.  Reviewer #2 raises several important points for you to consider, however, and I would encourage you to respond to these suggestions in a revised version of the article.

We look forward to receiving your revised manuscript.

Kind regards,

Joshua L Rosenbloom

Academic Editor

PLOS ONE

Journal Requirements:

Reviewers' comments:

Reviewer's Responses to Questions

**Comments to the Author**

1. Is the manuscript technically sound, and do the data support the conclusions?

Reviewer #1: Yes

Reviewer #2: Yes

2. Has the statistical analysis been performed appropriately and rigorously? 

Reviewer #1: Yes

Reviewer #2: Yes

3. Have the authors made all data underlying the findings in their manuscript fully available?

Reviewer #1: Yes

Reviewer #2: Yes

4. Is the manuscript presented in an intelligible fashion and written in standard English?

Reviewer #1: Yes

Reviewer #2: Yes

5. Review Comments to the Author

Reviewer #1: See attached file for more details. I think the paper is good and valuable. It adds something quite useful in fact. I highlight some minor changes to be made in a revised version. However, when I say "Revised", I do not mean a revise and resubmit, I mean minor modifications to be included in what I deem to be a valuable and publishable paper.

Reviewer #2: One of the puzzles that I was left with is how the marriage market was able to clear with such a large sex ratio. You note that in some cases, the sex ratio was as high as 14 but it didn't affect whether men eventually married. I think helping the reader through how that works would be really helpful. Your data is really great of distinguishing between tempo and quantum and it would be great to guid the reader through that reconciliation.

In the text, you note one explanation that: "For instance, men in disadvantageous marriage markets might have migrated to more promising marriage markets." This raises some issue about selection into the sample. You talk about avoiding selection into the sample by excluding immigrants from the analysis sample (but including them in the sex ratio). However, it wasn't clear how you were dealing with this other type of selection.

6. PLOS authors have the option to publish the peer review history of their article (what does this mean?). If published, this will include your full peer review and any attached files.

Reviewer #1: **Yes: **Vincent Geloso

Reviewer #2: No

---

## [Author Response · Author response to Decision Letter 0]

22 Mar 2022

The suggestions offered by both reviewers have been very helpful to improve our paper. We appreciate your effort to engage with our manuscript. We have included the reviewer comments below and respond to them individually.

1 Reviewer 1

1.1 Economic history 

The authors are, however, a bit careless in their historical details. We know that most males who migrated came as soldiers who were subsidized for the transatlantic travel and were offered incentives to stay behind after fighting the Iroquois (this is known as the Carignan-Sallieres regiment). However, the authors omit important sources of male migration { indentured servants, deported salt smugglers, wayward sons given lettres de cachet by the crown (Paul, 2008). Overall, some 30,000 people came to New France { but only 15,000 remained 1 (Boleda, 1990). As a share of the first number, Boleda (1990) places the share at 1/3 but probably a smaller share when one considers only those who stayed. 

Why is that an issue one might wonder? Because of how Moogk describes the migrants to New France: as reluctant migrants (Moogk, 1989). Reluctant because many were sent against their will in one way or another or had to be heavily subsidized to do so. Of all the migrants, only the indentured workers and a few others can be qualified as willing. Recent wage, income and price assembled evidence by Geloso (2019) and Geloso (2020) show that the inhabitants of New France were at best only modestly richer than the inhabitants of France (+20%) and at worst equally rich as the inhabitants of France. This suggest small to no gains to migration in contrast to the Spanish and British colonies where income gains for settlers were in excess of 2:1 relative to the respective mother countries. 

This causes a (minor) problem of interpretation because those who were deported/exiled/chosen by the government might have been selected on the basis of a factor that could be correlated with fertility decisions. For example, Galenson (1981) (for the American colonies), finds that a greater proportion of indentured servants migration meant more women coming of their own will. Indeed, the skewed sex ratio would entail a premium for women which would eventually reflect itself in the terms offered in indenture contracts. This premium would entice migration. 

In New France, a small share of indenture servitude would have hindered that mechanism. Now, notice I said it was a minor problem. That is because that concern does not alter (I do not see how) the results of the authors. However, it does speak to how they could extend their article to speak to two other literature: a) economic history; b) economics of immigration. I think that a small tweak to incorporate the elements above might allow the article to have a greater reach. Indeed, it may also even speak to points made by Gillian Hamilton on why Quebec had strange (and exceptionally strong) property rights regimes for women in marriage (compared to the rest of the Americas) (Hamilton, 1999). 

Thank you very much for these very insightful and helpful comments. We fully agree that these are important aspects of the historical circumstances and relevant to our paper. We incorporated your points into section 2.2 Historical context and observation period of our manuscript to better describe the circumstances and context of male immigration to the St Lawrence Valley. 

1.2 Age Gap Across Gender At Marriage 

I would also encourage the authors to try and provide contemporary comparison on the age gap at marriage. For example, it would be good to know what was the gap in France and the American colonies to the south of Quebec. I would recommend the works of Diebolt et al. (2017) on France as it contains a ton of statistical information for France from 1740 onwards. I would be satisfied with a comparison in 1740 for example. All I think is necessary is a comparison to highlight that Quebec was indeed exceptional. 

Thank you very much for the suggested reference. We agree that comparing France and the colony is indeed a very insightful aspect. We added a comparison between the ages at first marriage in France and the St. Lawrence Valley based on Diebolt et al. (2017). Mean ages at first marriage were indeed substantially lower in the St. Lawrence Valley colony than in France. In the colony median age for first marriage among women was around 22 and 26 for men in the 1740s. This is drastically lower than the 29 for men and 27 for women in France reported by Diebolt et al. (2017). Unfortunately, we haven’t found reference data for France covering the early part of our observation period. However, we suspect that the difference would be even larger, particularly for women. 

1.3 A short question

The authors claim that "Therefore, our findings do not support concerns that male-skewed sex ratio contexts necessarily result in an increase of never married men". I would be careful here as there is a question that lingers about male mortality. Many males were involved seasonally in the fur trade { a physically demanding but economically remunerative trade. If men died in sufficient numbers during altercations with the natives or as deaths during the trade, the results would be driven by this would it not? Moreover, men had militia duty of importance to fight off the natives. This meant also greater mortality. As such, the results could be driven by the overall level of violence as well? Is that a correct claim? Should it be added to the section on page 25?

We fully agree that our interpretation with respect to contemporary issues around male-skewed sex ratios deserves a little more contextualization. We amended the paragraph on page 25 to highlight the key role the high mortality context to shape the chance to marry for those men who survive more clearly. 

2 Reviewer 2

One of the puzzles that I was left with is how the marriage market was able to clear with such a large sex ratio. You note that in some cases, the sex ratio was as high as 14 but it didn't affect whether men eventually married. I think helping the reader through how that works would be really helpful. Your data is really great of distinguishing between tempo and quantum and it would be great to guid the reader through that reconciliation.

We acknowledge that we should better highlight the key role of the high mortality context to shape the chance to marry for those men who survive more clearly. In essence, men either married or died before reaching 45, thus resulting in a very small fraction of never married men at age 45. 

We amended the discussion of our results to better emphasize how the high mortality is the key factor for balancing out the sex ratio on the marriage market to generate no substantial cure fraction. 

In the text, you note one explanation that: "For instance, men in disadvantageous marriage markets might have migrated to more promising marriage markets." This raises some issue about selection into the sample. You talk about avoiding selection into the sample by excluding immigrants from the analysis sample (but including them in the sex ratio). However, it wasn't clear how you were dealing with this other type of selection.

We acknowledge that the quoted phrase did not clearly convey that it refers to a problem of other studies that rely on cross-sectional data. Given our semi-closed study population and the data covering the entire colony, we can avoid the problem of individuals finding a spouse outside of our observation scope. 

Restricting our sample to individuals born in the colony allows us to circumvent another source of unobserved emigration: immigrants who return to France after spending some time in the colony. Such re-migration is not an issue for those born in the colony. Re-migrants returning to France of course were part of the marriage market for the time they were present in colony and ideally, we would like them to be also represented in the sex ratio measures. However, these mostly male re-migrants left no trace in the vital statistics unless they married or had children in the colony. Thus, the true sex ratio might have been even more male-skewed, particularly during the early period.

Given that we analyze the marriage patterns for only those individuals born in Quebec, we avoid the issues of selection into the sample. All individuals in our sample are born in the colony and thus are restricted to the colonial marriage market.

---

## [Decision Letter · Decision Letter 1]

21 Apr 2022

Sex ratios and union formation in the historical population of the St. Lawrence Valley

PONE-D-21-25452R1

Dear Dr. Filser,

We’re pleased to inform you that your manuscript has been judged scientifically suitable for publication and will be formally accepted for publication once it meets all outstanding technical requirements.

Kind regards,

Joshua L Rosenbloom

Academic Editor

PLOS ONE

Additional Editor Comments (optional):

Reviewers' comments:

Reviewer's Responses to Questions

**Comments to the Author**

1. If the authors have adequately addressed your comments raised in a previous round of review and you feel that this manuscript is now acceptable for publication, you may indicate that here to bypass the “Comments to the Author” section, enter your conflict of interest statement in the “Confidential to Editor” section, and submit your "Accept" recommendation.

Reviewer #1: All comments have been addressed

Reviewer #2: All comments have been addressed

2. Is the manuscript technically sound, and do the data support the conclusions?

Reviewer #1: Yes

Reviewer #2: (No Response)

3. Has the statistical analysis been performed appropriately and rigorously? 

Reviewer #1: Yes

Reviewer #2: (No Response)

4. Have the authors made all data underlying the findings in their manuscript fully available?

Reviewer #1: Yes

Reviewer #2: (No Response)

5. Is the manuscript presented in an intelligible fashion and written in standard English?

Reviewer #1: Yes

Reviewer #2: (No Response)

6. Review Comments to the Author

Reviewer #1: I am very satisfied by the revised paper. Its very good and should be published as the authors answered my concerns and those of the other referee.

Reviewer #2: (No Response)

7. PLOS authors have the option to publish the peer review history of their article (what does this mean?). If published, this will include your full peer review and any attached files.

Reviewer #1: **Yes: **Vincent Geloso

Reviewer #2: No

---

## [Editor Report · Acceptance letter]

13 May 2022

PONE-D-21-25452R1 

Sex ratios and union formation in the historical population of the St. Lawrence Valley 

Dear Dr. Filser:

I'm pleased to inform you that your manuscript has been deemed suitable for publication in PLOS ONE. Congratulations! Your manuscript is now with our production department. 

Kind regards, 

on behalf of

Dr. Joshua L Rosenbloom 

Academic Editor

PLOS ONE